Spatio-temporal evolution and prediction of carbon storage in Kunming based on PLUS and InVEST models

Li Yimin 1 2
Yang Xue 1 yx9912130@163.com
Wu Bowen 1
Zhao Juanzhen 3
Jiang Wenxue 1
Feng Xianjie 3
Li Yuanting 3
1 School of Earth Sciences, Yunnan University , Kunming City, Yunnan , China
2 Yunnan Provincial University Domestic High Score Satellite Remote Sensing Geological Engineering Research Center , Kunming City, Yunnan , China
3 Institute of International Rivers and Ecological Security, Yunnan University , Kunming City, Yunnan , China
Meraj Gowhar
Electronic publication date: 2023 May 23
Publication date: 2023
Volume: 11
Electronic Location ID: e15285
Received 2023 Jan 3; Accepted 2023 Apr 3
Copyright: © 2023 Li et al.
Copyright year: 2023
Copyright holder: Li et al.
License: This is an open access article distributed under the terms of the Creative Commons Attribution License, which permits unrestricted use, distribution, reproduction and adaptation in any medium and for any purpose provided that it is properly attributed. For attribution, the original author(s), title, publication source (PeerJ) and either DOI or URL of the article must be cited.
License URL: https://creativecommons.org/licenses/by/4.0/

Keywords: InVEST model, Carbon storage, PLUS model, Kunming city, Spatial autocorrelation analysis

Funding: National Natural Science Foundation of China 41161070 Yunnan Science and Technology Department & Yunnan University Joint Fund Key Projects 2019FY003017 China Geological Survey Project DD20221824 China and Myanmar Ecological Conservation Joint Laboratory K26202000920 Innovation Team of Greater Mekong Subregion Climate Change Research Department, Yunnan University 2019HC027 All the external funding and sources of support received during this study came from the National Natural Science Foundation of China under Grant 41161070, the Yunnan Science and Technology Department & Yunnan University Joint Fund Key Projects (2019FY003017), the China Geological Survey Project (DD20221824), the China and Myanmar Ecological Conservation Joint Laboratory (K26202000920), and the Innovation Team of Greater Mekong Subregion Climate Change Research Department, Yunnan University (2019HC027). There was no additional external funding received for this study. The funders had no role in study design, data collection and analysis, decision to publish, or preparation of the manuscript.

==============================
Carbon storage is a critical ecosystem service provided by terrestrial environmental systems that can effectively reduce regional carbon emissions and is critical for achieving carbon neutrality and carbon peak. We conducted a study in Kunming and analyzed the land utilization data for 2000, 2010, and 2020. We assessed the features of land utilization conversion and forecasted land utilization under three development patterns in 2030 on the basis of the Patch-generating Land Use Simulation (PLUS) model. We used the Integrated Valuation of Ecosystem Services and Trade-offs (InVEST) model to estimate changes in carbon storage trends under three development scenarios in 2000, 2010, 2020, and 2030 and the impact of socioeconomic and natural factors on carbon storage. The results of the study indicated that (1) carbon storage is intimately associated with land utilization practices. Carbon storage in Kunming in 2000, 2010, and 2020 was 1.146 × 108 t, 1.139 × 108 t, and 1.120 × 108 t, respectively. During the 20 years, forest land decreased by 142.28 km2, and the decrease in forest land area caused a loss of carbon storage. (2) Carbon storage in 2030 was predicted to be 1.102 × 108 t, 1.136 × 108 t, and 1.105 × 108 t, respectively, under the trend continuation scenario, eco-friendly scenario, and comprehensive development scenario, indicating that implementing ecological protection and cultivated land protection measures can facilitate regional ecosystem carbon storage restoration. (3) Impervious surfaces and vegetation have the greatest influence on carbon storage for the study area. A spatial global and local negative correlation was found between impervious surface coverage and ecosystem carbon storage. A spatial global and local positive correlation was found between NDVI and ecosystem carbon storage. Therefore, ecological and farmland protection policies need to be strengthened, the expansion of impervious surfaces should be strictly controlled, and vegetation coverage should be improved.

Introduction

Due to rapid urbanization and industrialization, almost all cities around the world have experienced several climate-related and environmental problems (Cao, 2019; Sarkodie, Owusu & Leirvik, 2020), such as acid rain and the greenhouse effect, which are linked to the increasing intensity of land utilization by humans (Xu et al., 2018). Carbon dioxide has a significant impact on the climate. Greenhouse gases, such as carbon dioxide, emitted in unusually large quantities due to human activity, are the primary causes of global warming and exacerbate climate instability (Gao et al., 2022a; Yang, Liu & Bi, 2022; Zhang & Gu, 2022). Carbon dioxide can be stored in vegetation and soil, reducing the amount of CO2 in the atmosphere (Dorendorf et al., 2015). Carbon storage is critical for climate regulation and an important ecosystem service function. China is the world’s leading source of carbon emissions. At the United Nations General Assembly’s 75th session (2020), China pledged to reach the carbon emissions peak by 2030 and achieve carbon neutrality by 2060. (Chen et al., 2022a).

The emission of large amounts of greenhouse gases seriously threatens the global climate and environment. Several studies in the field of the ecological environment have estimated regional carbon storage and carbon emissions (Cai & Peng, 2021; Gogoi, Ahirwal & Sahoo, 2022; Li et al., 2022). Carbon storage refers to the amount of carbon stored in various forms. Traditional estimation methods and model methods are primarily used to estimate carbon storage. Traditional estimation methods include sample inventory (Li et al., 2021) and ecosystem carbon flux monitoring (Yang, Liu & Bi, 2022), and evaluation models include CASA (Tong, Zhang-Guo & Wei, 2016), Bookkeeping (Kong et al., 2018), and InVEST (Zhang et al., 2022a), among others. Due to its high workload and low efficiency, the traditional estimation method is only suitable for small-area carbon storage research. The InVEST model is broadly applicable to the field of carbon stock research (Chen et al., 2021; Wang, Gong & Deng, 2022; Zhang et al., 2022b; Zhao et al., 2022) because it demands a small amount of data, has high speed of operation, and performs convenient space-time visualization. Studying the current space-time characteristics of carbon storage and predicting future variations in land use and carbon storage can help realize the ‘double carbon’ goal. Several researchers have investigated prospective land utilization and carbon storage prediction at different scales. With the FLUS and InVEST models, Shao et al. (2022) predicted the evolution of Beijing’s carbon storage in 2035 under the natural evolution scenario, population evacuation urban development scenario, and green intensive ecological protection scenario. They also conducted zoning management studies based on spatial autocorrelation models. Li et al. (2020a) applied the SEUTH model to estimate the urban growth of Wuhan under different scenarios in 2030 and determined the consequences of urban sprawl on local carbon storage in combination with the InVEST model. Using the PLUS and InVEST models, Rukeya et al. (2022) dynamically evaluated the characteristics of land utilization and carbon storage varies in city clusters on the northern slope of the Tianshan Mountains under different scenarios from 2000 to 2030. Ding et al. (2022) utilized the PLUS and InVEST models to investigate and predict the evolution in land use and carbon storage around Hangzhou Bay since 2000 to 2018 and 2018 to 2030.

The models used to mimic futuristic land utilization/land cover mainly include CA-Markov (Song et al., 2022), FLUS (Xie et al., 2022), SD (Zhang et al., 2020b), PLUS (Yang, Su & Zhao, 2022), etc. Among them, the Patch-generating Land Use Simulation (PLUS) is a relatively new land utilization/land cover forecast model. By mining various driving factors, using the land expansion analysis strategy (LEAS), the random forest algorithm was used to obtain the development possibility of each category (Liang et al., 2021) to simulate the future changes in land use patches with greater accuracy in different years and different environments. The Markov model has higher accuracy in predicting quantities (Zhao et al., 2022). The integration of the PLUS model with the Markov model allows for a better estimation of regional land utilization/land cover in different future development scenarios.

Kunming has been a fast-growing city in western China for the past two decades; thus, it can be used as a representative of the rapid urbanization process. Kunming’s rich forest resources and biodiversity constitute a significant ecological barrier in China and even the world. Forest resources are closely related to carbon storage, so the rich forest resources in Kunming affect the changes in ecosystem carbon reserves, affecting the global carbon cycle. Further, the rich biodiversity of Kunming helps in maintaining the stability of the earth’s ecosystem and promote the sustainable development of human beings. The model method has been used in several studies to predict the space-time evolution of regional carbon storage. Therefore, in this study, we selected Kunming City in Yunnan Province as the study area, taking the three-period land use raster dataset and driving factors from 2000 to 2020 as the basis. The goals of this study were as follows: (1) Based on the PLUS model, the land-use pattern under different development patterns in Kunming City in 2030 was predicted, and the land utilization transformation trend from 2000 to 2030 was analyzed. (2) The InVEST model was adopted to assess the time-space distribution and change of carbon storage in Kunming from 2000 to 2030 and determine the influential effect of land use on carbon storage in different periods. (3) Analysis of spatial correlation and influencing factors of carbon storage in ecosystem.

Materials and Methods

Study area

Kunming is directly northeast of central Yunnan Province (102°10’ ~ 103°40’ E, 24°23’ ~ 26°22’ N) in the central Yunnan-Guizhou Plateau (Fig. 1) (Fang et al., 2021), the land area of Kunming is about 21,011.41 km2.

Figure 1 Study area.

Kunming, the capital of Yunnan Province in China, houses most of the province’s population and GDP. It is the province’s most important economic development center and corridor for economic and cultural communications between China and Southeast Asian countries. Furthermore, Kunming has abundant forest resources and biodiversity, making it an important area in the Yangtze River Economic Belt and an eco-conservation shield upstream of the Yangtze River (Jia, Wang & Zhao, 2022). In October 2021, the 15th Conference of the Parties (COP15) of the UN Convention on Biological Diversity was hosted in Kunming. At the conference, new ideas were presented for conserving worldwide biodiversity. Ecological civilization refers to the sum of material and spiritual achievements made by human beings in accordance with the objective law of harmonious development of man, nature, and society. During the 14th five-year plan period of China’s national economic and social development, Yunnan Province pledged that it would strive “to become the vanguard of China’s ecological civilization construction” as a long-term goal and incorporate the aim to achieve “carbon peak and carbon neutrality” into the general arrangement of economic development and the establishment of an eco-civilization. To realize these goals, Yunnan should promote the establishment of an eco-civilization, which is not only a significant embodiment of Yunnan’s initiative to serve the national developmental strategy but also an important embodiment of integrating into the national development. Therefore, studying and forecasting the response of land utilization conversions to carbon storage in Kunming can help build a strong ecological security barrier in southwest China and provide theoretical support for reducing regional emissions.

Data sources and preprocessing

The main data used for this research mainly included land utilization data, carbon density, driving factors, and other data (Table 1). (1) Land utilization data: the land-use data (30 m × 30 m) was obtained from the Resource and Environmental Science and Data Center of the Chinese Academy of Sciences. The data set was built by the Chinese Academy of Sciences through artificial visual interpretation on the basis of the national resources and environment database and the Landsat remote sensing image data as the main information source. The data were selected in time: 2000, 2010, and 2020, and the geographic coordinate system used was GCS_WGS_1984, which included 25 secondary land types and six primary land types (cultivated land, forest land, grassland, water area, construction land, and unused land). (2) Carbon density: the carbon density varies by land use type. The studies which had similar levels of natural resources (Ke & Tang, 2019) and climatic conditions (Yan et al., 2015; Tang, Xu & Ai, 2019) to that in the study area were selected as references. Then, the data were corrected according to the above-ground carbon density dataset of the terrestrial ecosystem of China in 2010 and the carbon density dataset of soil in 0 ~ 100 cm of the terrestrial ecosystem of China in 2010. Finally, the carbon density dataset for land utilization categories in Kunming was obtained (Table 2). (3) Driving factors: The driving factor data for future land prediction included physical factors and social factors. The physical factors involved the DEM, slope, annual average rainfall, soil type, annual average temperature, and distance to the water system. The social factors included the GDP, distance to the government office, distance to the highway, distance to the main road (first-class road and second-class road), population density, data source, and resolution, as shown in Table 1. Next, ArcGIS was used to unify the resolution to 30 m × 30 m and unified the geographic coordinate system to GCS_WGS_1984. (4) Other data: four Landsat 8 OLI and TIRS images in August 2020 were selected for radiometric calibration, atmospheric correction, splicing, and cropping to obtain remote sensing images of Kunming City to extract the impervious surface in 2020. NDVI was directly obtained from the vegetation index data MOD13A1.

Table 1 Data and sources.

Data type	Data description	Resolution/m	Data source	
Land use data	2000 to 2020	30	Resource and Environment Science and Data Center, Chinese Academy of Sciences (https://www.resdc.cn/)	
Carbon density	–	–	Scientific literature statistics, 2010 Chinese terrestrial ecosystem aboveground carbon density data set, 2010 Chinese terrestrial ecosystem soil 0~100 cm carbon density data set.	
Other data	Landsat 8 OLI and TIRS	30	United States Geological Survey Official Website (https://earthexplorer.usgs.gov/)	
MOD13A1	500	NASA official website (https://search.earthdata.nasa.gov/search)	
Driving factors	GDP	1,000	Resource and Environment Science and Data Center, Chinese Academy of Sciences (https://www.resdc.cn/)	
Population density	1,000	
Distance to government office	30	
Distance to main road	30	Openstreetmap (https://www.openstreetmap.org/)	
Distance to highway	30	
Elevation	30	Geospatial Data Cloud Official Website (https//www.gscloud.cn/)	
Slope	30	
Distance to water system	30	Resource and Environment Science and Data Center, Chinese Academy of Sciences (https://www.resdc.cn/)	
Mean annual temperature	1,000	
Mean annual rainfall	1,000	
Soil type	1,000	

Table 2 Carbon density database of land use types in Kunming.

Land use	Aboveground carbon density	Underground carbon density	Soil carbon density	Carbon density of dead organic matter	
Cultivated land	1.31	0.73	11.65	0	
Forest land	40.41	10.45	42.75	2.62	
Grassland	2.55	8.31	15.1	0.85	
Water area	0	0	0	0	
Construction land	0	0	0	0	
Unused land	0	0	4.2	0	

Methods

The research framework was composed of four stages (Fig. 2). (1) Data preparation and preprocessing. (2) The PLUS model predicted land utilization changes for 2030 under the trend continuation scenario, the eco-friendly scenario, and the comprehensive development scenario. (3) Using the InVEST model, the space-time allocation of carbon storage under various development scenarios from 2000 to 2030 was evaluated. The trend of land use revolution in the study area from 2000 to 2030 was discovered to determine the impact of land utilization variation on carbon sink. (4) The spatial correlation of carbon storage was evaluated using the spatial autocorrelation model, and the influencing factors of carbon storage were analyzed.

Figure 2 Research framework.

Ecosystem carbon storage estimation

The Integrated Valuation of Ecosystem Services and Trade-offs (InVEST) model is an ecological model for evaluating and quantifying ecosystem services. In this study, the carbon storage and sequestration module of the InVEST model was used to calculate the carbon storage in Kunming. The basic assumption of this module is that the value of the carbon density of a specific land category is fixed, and the carbon storage of that land type can be acquired by multiplying the value of carbon density with that of the land area (Gong et al., 2022). The carbon storage module in the InVEST model divides the ecosystem carbon storage into four basic carbon pools (Lin et al., 2022), which include terrestrial biogenic carbon, subsurface biogenic carbon, soil carbon, and dead organic carbon. The sum of the carbon stock of the four carbon pools provides the value for the total carbon storage of the ecosystem in the area, which can be achieved using Eqs. (1) and (2) (Li et al., 2020b).

(1) Ci=Ci−above+Ci−below+Ci−soil+Ci−dead

(2) Ctot=∑i=1n⁡Ci×Si.

Here, Ci indicates the carbon density of land utilization type i; Ci−above, Ci−below, Ci−soil, and Ci−dead indicate the carbon density of terrestrial biogenic carbon, subsurface biogenic carbon, soil carbon, and dead organic carbon of land use type i, respectively; Ctot indicates the total carbon storage in the region; Si indicates the area of land utilization pattern i; n indicates the total number of land use types.

Future land use prediction and scenario setting

PLUS model

In this study, the PLUS model was utilized to forecast land utilization in Kunming under distinct growth scenarios in 2030. Patch-generating Land Use Simulation (PLUS) is a new land use/land cover prediction model proposed by Liang et al. (2021) (China University of Geosciences). Compared to other traditional prediction models, it has improved emulation capability and can measure the landscape pattern more precisely (Yang et al. 2022). The PLUS model mainly combines the land expansion analysis strategy (LEAS) and the CA model with multi-type random patch seeds (CARS). The LEAS module is employed to extract the land cover expansion. Then, the random forest classification algorithm is utilized to mine the probability of change and inertia of each land use type (Gao et al., 2022b). The CARS module affects the local land contestation process through a self-adaptive coefficient and drives the changes in land utilization intensity to meet the upcoming land use demand. For forecasting, the number of prospective land application patches, a Markov model with high simulation accuracy is selected (Sun & Liang, 2021).

Scenario setting

The development planning and the land use change of Kunming City are affected by many factors. The previous planning of Kunming City started in 2006 and ended in 2020. The next overall planning of land and space in Kunming City started in 2021 and is planned till 2035; 2030 is the intermediate node of the next land and space planning of Kunming City. Additionally, following the implementation outline of “Kunming City building regional international center city (2017–2030)” issued in September 2017, Kunming City will be fully transformed into a regional international center city in southwest China in 2030 (Zhang, 2019). Therefore, in this study, we selected 2030 as the prediction year of future land use in Kunming.

Following the historical law of land utilization change in Kunming, along with the development situation and future planning, and assuming that the region can meet the future natural, social, and economic needs, the PLUS model was used to construct three scenarios (Cui et al., 2022; Ding et al., 2021; Zhang & Gu, 2022) for the coming land utilization expansion of Kunming. These scenarios are as follows:

1) Trend continuation scenario (S1). According to the law of land utilization change in Kunming between 2010 and 2020, only the water area can be controlled as the restricted conversion area, and the construction land cannot be transferred to alternative land types. Markov model is a method to simulate or predict the occurrence probability of events based on the transition probability matrix. Using the Markov model, the land use types of Kunming in the 2030 trend continuation scenario were predicted.

2) Ecological protection scenario (S2). The Yunnan Province is committed to leading the construction goals of ecological civilization, guided by ecological and environmental protection, by limiting the large extension of construction land, increasing the shift of other types of land to woodland and grassland, and reducing the conversion of eco-land to unused land.

3) Comprehensive development scenario (S3). While ensuring economic development, the protection of the ecosystem, substantially reducing the transfer of non-construction land to construction land, and increasing the transfer probability of unused land to ecological land without affecting economic development need to be considered. Since cultivated land is economically important, the conservation policy of cultivated land needs to be considered to strictly control the transformation of cultivated land.

The neighborhood weight reflects the expandability of a certain land utilization type. When the value is closer to one, the extension capacity of the land utilization type is stronger. Following the law of land utilization conversion in Kunming for the period 2010–2020 and based on the findings of published studies (La et al., 2021), the neighborhood weights of each land utilization type under various scenarios were set by comparing the accuracy of the outcomes of simulation based on different parameters (Table 3).

Table 3 Neighborhood weights of simulation scenarios.

Scenarios	Cultivated land	Forest	Grassland	Water body	Construction land	Unused land	
Trend continuation	0.50	0.77	0.50	0.58	0.83	0.50	
Ecological protection	0.50	0.82	0.75	0.58	0.70	0.60	
Comprehensive development	0.50	0.77	0.75	0.58	0.77	0.60	

Verifying the accuracy of the simulation

According to the land use data of Kunming City in 2010, the PLUS model was employed to forecast the results of the land utilization type of Kunming City in 2020. The Kappa coefficient and overall accuracy (OA) were used to evaluate simulation accuracy, in which the Kappa coefficient is a consistency checking method based on a confusion matrix. By comparing the simulated results with the real land use data in 2020, the Kappa coefficient of the two sets of data was 0.8638, and the overall accuracy (OA) was 90.72%. The simulation accuracy was high, indicating that the model and various parameters might be used for simulating the land use of Kunming City in the future.

Impervious surface extraction

The impervious surface area (ISA) is a typical land cover type and an important indicator for measuring and analyzing city development and ecology (Xu, 2009). The index method is generally adopted to extract the urban impervious surface. The representative spectral indices include the normalized building index (NDBI), the normalized difference impervious surface index (NDISI), and the enhanced normalized difference impervious surface index (ENDISI) (Duan, Zhang & Liu, 2022). ENDISI can better identify shadows of mountains and remains unaffected by terrain factors while identifying impervious surfaces. Kunming City is dominated by mountainous terrain. Selecting this index for extracting information on impervious surfaces can prevent mountain shadows from affecting the extraction accuracy of impervious surfaces. The ENDISI can be calculated using Eq. (3).

(3) ENDISI=(2Blue+MIR2)÷2−(NIR+Red+MIR1)(2Blue+MIR2)÷2+(NIR+Red+MIR1).

Here, Blue, Red, NIR, MIR1, and MIR2 are the reflectance of blue, red, near-infrared, shortwave infrared 1, and shortwave infrared, two bands corresponding to the image.

Before extracting the information on the impervious surface, the normalized water index MNDW (Xu, 2008) needs to be used to mask the large area of water and snow over the study area. The MNDW index can be derived from Eq. (4).

(4) MNDWI=(Green−MIR1)(Green+MIR1).

Here, Green, MIR1, and MIR2 are the reflectance of green, shortwave infrared 1, and shortwave infrared 2 bands corresponding to the image, respectively.

Using ArcGIS, 700 verification points were randomly generated in the working area after masking water and snow. With the data on the impervious surface extracted from the Landsat 8 image in 2020, the land use data and the remote sensing imagery of Kunming City in this period were selected, and the accuracy of the extracted impervious surface was determined by artificial visual interpretation. The Kappa coefficient was 0.7582, and the overall accuracy was 89.86%. It demonstrated that the accuracy of the impervious surface met the requirements of subsequent research.

Results

Land-use change analysis

According to the overlay map of the area proportion of land utilization types (Fig. 3), the land in Kunming City between 2000 and 2020 was primarily forest land, accounting for approximately 45% of the total area studied. The total area of forest land, grassland, and cultivated land decreased between 2000 and 2020. From 2000 to 2020, the land use transfer matrix (Table 4) revealed that forest land decreased by 142.28 km2, grassland decreased by 328.95 km2, and cultivated land decreased by 278.07 km2. The growth rate of construction land was high; construction land increased by 708.84 km2, and the area proportion increased from 2.3% to 5.7%. The development process of S1 in 2030 was similar to the trend of 2000–2020. Woodland, grassland, and cultivated land in 2030 decreased by 123.86 km2, 150.61 km2, and 197.358 km2, respectively. However, construction land increased by 451.136 km2, and water area and unused land remained unchanged. In the S2 scenario, with ecological protection as the leading role, the area of forest land increased by 1.1%, and the growth rate of construction land slowed down. In the S3 scenario, where cultivated land conservation policy and economic development were considered, cultivated land and construction land increased by 55.82 km2 and 209.28 km2, respectively. In comparison, forest land and grassland decreased by 123.96 km2 and 151.79 km2, respectively.

Figure 3 The overlay map of the area proportion of land utilization in Kunming from 2000 to 2030.

Table 4 Land utilization transfer matrix of Kunming from 2000 to 2020.

2020	
Land use	Cultivated land	Forest land	Grassland	Water area	Construction land	Unused land	Total	
2000	Cultivated land	3,426.40	218.07	196.14	25.74	387.23	0.80	4,254.38	
Forest land	230.43	8,765.48	441.07	40.69	119.74	1.79	9,599.20	
Grassland	268.31	456.62	5,154.64	19.52	236.11	6.52	6,141.72	
Water area	17.92	4.07	6.27	420.81	24.26	0.32	473.65	
Construction land	32.26	11.51	10.47	4.20	425.58	0.19	484.21	
Unused land	0.99	1.17	4.18	0.51	0.13	60.91	67.89	
Total	3,976.31	9,456.92	5,812.77	511.47	1,193.05	70.53	21,021.05	

The construction land in the four periods was concentrated in the main urban area of Kunming (Wuhua District, Panlong District, Xishan District, Guandu District, and Chenggong District) and the surrounding areas (eastern Anning City, northern Songming County, and northern Jinning District) (Fig. 4). According to the statistical yearbook of Yunnan Province in 2020, the main urban area of Kunming comprised 63.17% of the population and generated 73.76% of the GDP of the city. A large population and capital flow led to increased construction, and ecological lands, such as forest land and grassland, decreased considerably. The land utilization types in Luquan County and Xundian County are mainly woodland and grassland, which act as important ecological barriers in Kunming City. The Jiaozi Snow Mountain Nature Reserve is in Luquan County and is an important water conservation ecological reserve in Kunming City. The overall territorial spatial layout of Kunming is: the southern Dianchi Lake basin is the core of the economy and population, and the northern mountainous area is the ecological security area.

Figure 4 (A–F) Land use changes in Kunming from 2000 to 2030.

Analysis of space and time variation of ecosystem carbon storage

The carbon storage in Kunming in 2000, 2010, and 2020 was 1.146 × 108 t, 1.139 × 108 t, and 1.120 × 108 t, respectively, indicating a continuous decrease, with a total decrease of 2.619 × 106 t in 20 years. The reduction of carbon storage in 2010–2020 was the highest, with a decrease of 1.917 × 106 t. During this period, the economic development and urbanization of Kunming were rapid, and the demand for land use was relatively high.

In 2030, carbon storage is predicted to be 1.102 × 108 t in the S1, and the loss of ecosystem carbon stock is greater. The ecosystem carbon storage might decrease by 1.818 × 106 t compared to that in 2020. In 2030, carbon storage is predicted to be 1.136 × 108 t in the S2 scenario, which is 1.601 × 106 t higher than that in 2020, indicating that ecological protection can restore ecosystem carbon storage in Kunming. Finally, carbon storage in the S3 is predicted to be 1.105 × 108 t in 2030, and the ecosystem carbon storage might decrease by 1.479 × 106 t compared to that in 2020, which is less than the carbon loss in the S1. That is, the prediction results of ecosystem carbon storage under different scenarios are S2 > S3 > S1, which is consistent with the prediction results of previous studies (He et al., 2022; Wei et al., 2023). As shown in Fig. 5, the terrestrial biogenic carbon, subsurface biogenic carbon, dead organic carbon, and soil carbon in the S2 scenario in 2030 are predicted to be higher than those in the S1 and S3 scenarios. The soil carbon gap was found to be the largest, which was 1.539 × 106 t and 1.250 × 106 t higher than the soil carbon in the S1 and S3 scenarios, respectively. The terrestrial biogenic carbon storage, belowground carbon storage, and dead organic carbon storage in the S1 and S3 scenarios were similar. In contrast, the soil carbon storage in S3 was higher than that in S1, which was consistent with the changes in total carbon storage.

Figure 5 Changes of carbon stock in Kunming under three development scenarios in 2030.

The overall content of ecosystem carbon stock in Kunming is high, showing a ‘north high, south low’ distribution (Figs. 6 and 7). Carbon storage is primarily concentrated in Luquan County, Xundian County, and Yiliang County to the east and west of the research area, respectively, whereas the main urban area of Kunming City to the south of the study area, is highly urbanized and has less carbon stored. From 2000 to 2030, the main urban area of Kunming experienced the greatest change in the spatial layout of carbon storage. According to our findings using the S1 scenario, construction land in the main urban area expanded dramatically from 2000 to 2030, resulting in a loss of regional ecosystem carbon storage every year due to rapid urbanization. Under the S2 scenario’s ecologically protective development, the carbon stock in Kunming’s main urban area increased in 2030 compared to 2020. When considering the cultivated land conservation policy, eco-friendly policy, and economic development, carbon storage in the main urban area of Kunming was lower than that in 2020, but the reduction range was narrower than that in the S1 scenario. The results showed that carbon stock in the main urban area of Kunming could strongly affect the change in the ecosystem carbon storage in the whole city.

Figure 6 Changes in carbon storage in Kunming from 2000 to 2030.

Figure 7 (A–F) Spatial distribution of carbon storage in Kunming from 2000 to 2030.

Effects of land-use conversion on carbon storage

Land utilization/land cover significantly affects vegetation cover and biomass. It is also the primary purpose for distributing and changing carbon stock in regional terrestrial ecosystems (Zhang et al., 2022b). The dynamic changes in carbon stock induced by the changes in main land use types in Kunming from 2000 to 2030 are shown in Fig. 8. From 2000 to 2030, according to the S1 scenario, carbon storage decreased mainly because of the shift from forest land and grassland to other land use types. Although the policy of returning farmland to the forest has partly promoted the transfer of cultivated land to forest land, the expansion of construction land encroaches on forest land, leading to a sharp decrease in the ecosystem carbon reserve. Under the S2 scenario, in 2030, the increase in ecological lands, such as forest land and grassland, was predicted to be the key reason for the increase in carbon stock. Under the S3 comprehensive development scenario, in 2030, the cultivated land conservation policy promoted increased cultivated land carbon storage, while the forest land and grassland carbon stock decreased. The carbon density of forest land and grassland was higher than that of cultivated land, resulting in a decrease in the overall ecosystem carbon storage. The outcomes suggested that the conversion in land utilization type is consistent with the changes in the ecosystem carbon stock, and land use/land cover directly affects ecosystem carbon stock.

Figure 8 Changes of main land types and carbon storage in Kunming from 2000 to 2030.

Spatial correlation analysis of ecosystem carbon storage

Spatial correlation is segmented into spatial global autocorrelation and spatial local autocorrelation (Luo, Ai & Jia, 2022). In this study, Moran’s I index was employed to represent the spatial global autocorrelation of ecosystem carbon storage in Kunming (Xiong et al., 2021) and Getis-Ord Gi * was employed to measure the spatial local autocorrelation (Zhang et al., 2020a). This study divided Kunming into a 2 km × 2 km grid, and the carbon storage data in the three development scenarios for 2000–2020 and 2030 were linked to the grid. Based on this scale, the spatial correlation of carbon stock in Kunming was analyzed.

In terms of global autocorrelation, the spatial Moran’s I values of carbon storage in Kunming were 0.5583 in 2000, 0.5546 in 2010, 0.5635 in 2020, 0.5900 in 2030 S1, 0.5672 in 2030 S2, and 0.5763 in 2030 S3, indicating a significant spatial global autocorrelation in carbon storage in Kunming (all values were greater than 0). The results of the carbon storage hotspot analysis under the three development scenarios in Kunming from 2000 to 2030 are shown in Fig. 9 for local autocorrelation. From 2000 to 2020, the area designated as the hotspot of carbon stock in Kunming reduced, while the area considered to be a coldspot increased. Except for the northeast of Dongchuan District, the coldspot area in other areas, especially in the main urban area of Kunming, increased. In the last 20 years, the hotspots of carbon stock in the research area were scattered in the northwestern, western, central, and eastern regions, including the east and west sides of Luquan County, Xishan District, the western part of Songming County, the northern part of Yiliang County, and the eastern part of Guandu District. Carbon storage was not only higher in these areas, but high-value areas of carbon stock were also gathered in these areas, primarily situated in zones with less construction land, excellent plant coverage, and more ecological land. The coldspot area was mostly distributed in the northeastern, central, and southern locations of Kunming, i.e., the northeastern part of Dongchuan District, the eastern part of Songming County, the eastern part of Yiliang County, the northern and western parts of Shilin County, and the central part of the main urban area of Kunming City. These areas underwent rapid land development, had complex topography and geomorphology, and fragmented distribution of ecological land, forming an area with low carbon storage. In the S1 scenario, in 2030, the coldspot area in the main urban areas of Anning City and Kunming City increased. In the main urban area, the coldspot area in the S2 scenario was slightly smaller than in 2020. In the S3 scenario, the coldspot area in 2030 was similar to that in 2020, indicating that eco-friendly and cultivated land protection policies were beneficial in reducing the loss of ecosystem carbon stock in Kunming City.

Figure 9 (A–F) Getis-Ord Gi * analysis of carbon storage in Kunming from 2000 to 2030.

Research on driving factors of ecosystem carbon storage

Correlation between ecosystem carbon storage and impact factors

Based on the spatial-temporal allocation and variation of carbon stock in Kunming, in this study, we performed correlation analysis (Chen et al., 2022b; Zhang & Gu, 2022) to determine the effects of natural factors and socioeconomic factors on carbon storage. Taking the data on impact factors and carbon storage in 2020 as an example, the data were set to a grid of 2 km × 2 km according to the study area using the grid method. In total, 5,153 grid points were generated. The impact factors included impervious surface coverage, GDP, population density, altitude, NDVI, and annual average precipitation.

The results of Pearson’s correlation coefficient between carbon storage and various influencing factors in Kunming are shown in Table 5. Carbon storage was moderately negative correlated with impervious surface coverage, weakly negatively correlated with GDP and population density, strongly positively correlated with NDVI, and weakly positively correlated with altitude and annual rainfall. The results showed that at a grid scale of 2 km × 2 km, impervious surface and vegetation had the greatest impact on carbon storage in Kunming. Carbon storage was negatively correlated with impervious surface coverage and positively correlated with NDVI.

Table 5 Pearson correlation coefficient between carbon storage and impact factors in Kunming.

Impact factors	Impervious surface percentage	GDP	Population density	Elevation	NDVI	Average annual rainfall	
Carbon storage	−0.530	−0.2	−0.215	0.339	0.689	0.225	

Bivariate spatial autocorrelation analysis of carbon storage and the influencing factors

Depending on the results of Pearson’s correlation analysis, two indicators that had the strongest relevance with the carbon storage of the ecosystem in Kunming were selected, i.e., impervious surface coverage and NDVI, and bivariate spatial autocorrelation analyses were performed between the two indicators, and carbon storage data. The bivariate ‘Moran’s I index represents bivariate global spatial autocorrelation. A total of −0.379 was the bivariate ‘Moran’s I index of impervious surface coverage and ecosystem carbon storage. The ‘Moran’s I index’s negative value indicated a global negative correlation between impervious surface coverage and ecosystem carbon storage. The bivariate NDVI and ecosystem carbon storage index, Moran’s I, was 0.450. The presence of a positive ‘Moran’s I index value indicated a global positive correlation between NDVI and ecosystem carbon storage.

Figure 10a depicts the bivariate LISA cluster map of impervious surface coverage and carbon storage in the Kunming ecosystem, which reflects the two’s local agglomeration characteristics in space. The impervious surface coverage and carbon storage of the ecosystem showed opposite values of the two-pole agglomeration characteristics in space, which mainly were high-low agglomeration and low-high agglomeration, i.e., carbon stock in the area with low impervious surface coverage was higher, and carbon stock in the area with high impervious surface coverage was lower. The bivariate LISA clustering results of NDVI and ecosystem carbon storage in Kunming (Fig. 10b) showed that NDVI and ecosystem carbon storage had a similar value for the polar agglomeration characteristics in space, mainly high-high agglomeration and low-low agglomeration. Thus, the regional ecosystem carbon storage with higher vegetation coverage was higher, and the regional ecosystem carbon storage with lower value coverage was lower. These results suggested a local negative correlation between impervious surface coverage and ecosystem carbon storage and a local positive correlation between NDVI and ecosystem carbon storage.

Figure 10 Bivariate LISA cluster map of carbon storage and influencing factors in Kunming.

(A) Bivariate LISA cluster map of impervious surface coverage and carbon storage. (B) Bivariate LISA cluster map of NDVI and carbon storage.

Discussion

Contribution of land use driving factors

Social and natural factors mainly drive land use change (Gong et al., 2022). Social factors include population density, GDP, distance to road, distance to the government office, etc. Natural factors include terrain factors, such as slope and elevation, and climatic factors, such as mean annual rainfall and mean annual temperature. In this study, 11 driving factors of social economy and climate were selected to forecast land utilization in 2030, and the contribution of each driving factor was evaluated (Table 6). A higher contribution of a factor indicated a greater impact of the driving factor on local land use evolution.

Table 6 Contribution of driving factors of land use.

Driving factors	Land use	
	Cultivated land	Forest land	Grassland	Water area	Construction land	Unutilized land	
DEM	0.111	0.183	0.1	0.136	0.085	0.111	
GDP	0.166	0.091	0.133	0.187	0.143	0.166	
Distance to government	0.093	0.091	0.104	0.022	0.104	0.093	
Distance to highway	0.091	0.075	0.092	0.028	0.089	0.091	
Population density	0.104	0.125	0.117	0.168	0.154	0.104	
Average annual rainfall	0.089	0.081	0.089	0.069	0.111	0.089	
Distance to main road	0.089	0.063	0.08	0.017	0.063	0.089	
Distance to water system	0.079	0.071	0.104	0.097	0.069	0.079	
Slope	0.078	0.117	0.084	0.2	0.096	0.078	
Soil type	0.022	0.018	0.015	0.046	0.023	0.022	
Mean annual temperature	0.079	0.084	0.083	0.031	0.122	0.079	

The driving factors with the highest contribution to cultivated land were GDP (0.166), DEM (0.111), and population density (0.104). Cultivated land expansion is inextricably linked to population and economic development because rapid population and economic growth will lead to greater food demand (Liao et al., 2021). Climate conditions differed depending on altitude. Agricultural farming was closely linked to climatic conditions, so cultivated lands were generally distributed in low-altitude farming areas. DEM (0.183), population density (0.125), and slope were the factors that had the greatest impact on forest land (0.117). The slope and altitude were important topographic factors influencing vegetation growth. In general, areas with a small slope and a low altitude are better suited for vegetation growth. Higher population density areas have less forest land area, and population density limits forest land expansion. GDP (0.133), population density (0.117), and distance to the water system (0.104) were the driving factors with the greatest contribution to grassland. The GDP and population density restricted grassland expansion, while the water system promoted grassland expansion. The area closer to the water system was more prone to grassland expansion. The driving factors with the highest contribution to construction land were population density (0.154), GDP (0.143), and the mean annual temperature (0.122). A greater population density and a higher GDP were associated with a greater demand for and more expansion of construction land. The mean annual temperature affected the development of construction land by changing the population density. The driving factors with the highest contribution to the unused land were DEM (0.306), the mean annual temperature (0.199), and the mean annual rainfall (0.156). The climate in high-altitude areas is harsh, and the perennial temperature is low, which is unfavorable for vegetation growth and human habitation.

Suggestions on carbon sink function restoration and governance

The period from 2000 to 2020 was an important stage of economic development in Kunming. In the last 20 years, the gross domestic product increased from 6,262.853 to 67,337.909 million yuan, the population increased from 4.8094 to 8.463 million, and the construction land expanded from 484.21 to 1,193.05 km2. In the last 20 years, the area of forest land and grassland has decreased significantly, resulting in a decrease in ecosystem carbon storage in Kunming. Taking timely measures to protect the ecosystem is necessary to reduce the loss of carbon reserves, according to the predictions of the trend continuation scene, ecological protection scene, and comprehensive development scenario. According to the “14th Five-Year Plan for National Economic and Social Development of Yunnan Province and the Outline of the Visionary Goals for the Year 2035”, Yunnan Province focused on “making new progress in the building of eco-civilization by 2025, and building the vanguard of China’s ecological civilization by 2035” (People’s Government of Yunnan Province, 2021). Responding to climate change and achieving carbon peak and carbon neutrality are also necessary for future development. Kunming needs to incorporate the dual-carbon goal into the overall future development and foster a comprehensive greener shift in environmental development.

Based on the distribution of carbon stock in Kunming from 2000 to 2030 (Fig. 7), Luquan County, Xundian County, and Dongchuan District in the north and Yiliang County and Shilin County in the southeast were found to be the essential sources of carbon storage in Kunming. Luquan County is a high-altitude mountain area; the mountain area takes up 98.4% of the whole area of the county, the mountain slope is large, and the forest coverage rate is around 55.4%. Considering that Luquan County is dominated by agricultural development, based on these advantages, forestry, agricultural development, and ecological green development can be organically combined to carry out characteristic economic forests, fruit trees, and other afforestation. Focusing on the positioning of the ecological containment function area of Kunming, Luquan County, Xundian County, and Dongchuan District needs to strengthen ecological conservation construction and improve the protection system of nature reserves and forest parks. Strive to build the Jiaozi Snow Mountain Nature Reserve as an important water source conservation area and ecological protection guard not only in Kunming but also in the middle and upper reaches of the Yangtze River. These areas should build a strong ecological security defense line in the north of Kunming. As an ‘ecological civilization county of Yunnan Province’, Shilin County should lead the construction of the national ecological civilization and help Yunnan in achieving its target of ‘double carbon’. Considering that the tourism industry is important in Shilin County, eco-tourism should be promoted to construct an ecological civilization. Shilin County, as a typical karst area, should actively solve the problem of restoration of ecological vegetation and strengthen the implementation of comprehensive control of returning farmland to forest and rocky desertification. Yiliang County has an abundance of forest resources and species diversity. The forest coverage rate of the county is 46.2%. Yiliang County can utilize its resource advantages, strengthen the protection of forest resources based on the current condition of forestry resources. Through the construction of an economic forest and flower planting base, simultaneously develop an ecological civilization and forestry economy and build an ecological security barrier with Shilin County in the southeast of Kunming. During the period of 2000 to 2030, the carbon storage in the main urban area of Kunming City, especially in the Dianchi Lake Basin, was found to decrease every year. Promoting the ecological management of the Dianchi Lake Basin is imperative for the development of Kunming City. The main urban area of Kunming City is located in the red line of water conservation and eco-protection of plateau lakes and the upper reaches of the Niulan River. Thus, its resource advantages need to be fully utilized to drive the holistic and sustainable development of the plateau lake basin. It is also necessary to improve the ‘three lines and three zones’ delineation standards, strictly control the Dianchi Lake Basin’s wetland parks and basic farmlands, conduct rational land and space planning, and improve the efficiency of land resource utilization. Ways to meet the needs of urban expansion while also protecting the environment must be determined urgently for the development of Kunming City. Solving this problem is also the key to achieving the dual carbon goals and the establishment of eco-civilization in Yunnan Province.

Research prospect

With the PLUS and InVEST models, the space-time changes in carbon storage in Kunming City were predicted and evaluated. The influence of land utilization conversion on carbon stock was analyzed. The correlation between impervious surface coverage and vegetation index, and regional ecosystem carbon storage was analyzed. The findings provided new ideas for sustainable development in the future. This study, however, had some limitations. First, the InVEST model’s carbon storage calculation module had some flaws. Many factors influence the ecosystem’s carbon stock. The InVEST model ignored the effects of climate, topography, hydrology, and other conditions in favor of focusing solely on land use change. Furthermore, the model ignored the effect of interannual changes in carbon density. Second, when using the PLUS model for future land cover, only 11 driving factors were considered; however, the actual land utilization evolution is influenced by a large number of physical and human factors. In future studies, the measured data might be used to determine the dynamic carbon density. Besides land use, other influencing factors might be considered to comprehensively evaluate the regional ecosystem carbon storage. While performing land-use simulations, we should select as many driving factors as possible to predict the future requirement of land utilization and improve the prediction accuracy of future land utilization patterns.

Conclusions

Based on the data on land use and driving factors from 2000 to 2020, the PLUS and InVEST models were used to monitor and study the time-space dynamics of land and ecosystem carbon storage in Kunming from 2000 to 2030. The findings illustrated the following: (1) From 2000 to 2020, the land-use types in Kunming were mainly forest land. The total area of forest land, grassland, and cultivated land decreased, and construction land increased. In the S1, the construction land was predicted to expand greatly in 2030. In S2, the forest land area was predicted to increase greatly. The S3 predicted that cultivated land area would increase slightly, construction land would increase moderately, and forest and grassland would decrease slightly. (2) Carbon storage in Kunming was found to be ‘high in the north and low in the south’. The carbon storage of the ecosystem was found to be 1.146 × 108 t, 1.139 × 108 t, and 1.120 × 108 t in 2000, 2010, and 2020, respectively, indicating a continuous decrease. Carbon storage is expected to be 1.102 × 108 t in S1, 1.136 × 108 t in S2, and 1.105 × 108 t in S3 by 2030. (3) Kunming City has a significant spatial autocorrelation of carbon storage. In the local space, the ecosystem carbon storage in Kunming City from 2000 to 2020 was mainly characterized by the expansion of the coldspot area. In 2030, the coldspot area was predicted to increase in the S1 scenario, slightly decrease in the S2 scenario, and be similar to 2020 in the S3 scenario. (4) A global and local negative correlation was found between impervious surface coverage and ecosystem carbon storage. A global and local positive correlation was found between NDVI and ecosystem carbon storage. (5) The carbon reserves calculation module of InVEST model has some limitations. The influence of other driving factors was ignored while predicting future land use. Therefore, in future research, the measured data can be used to determine the dynamic carbon density, and the effects of other factors, except for land use, on the carbon density can be considered. When simulating future land use, as many driving factors as possible should be selected to simulate the future land use demand.

The authors gratefully acknowledge the support of their families and teachers in conducting this comprehensive study.

Additional Information and Declarations

Competing Interests

Author Contributions

Data Availability

The authors declare that they have no competing interests.

Yimin Li conceived and designed the experiments, analyzed the data, authored or reviewed drafts of the article, and approved the final draft.

Xue Yang conceived and designed the experiments, performed the experiments, analyzed the data, authored or reviewed drafts of the article, and approved the final draft.

Bowen Wu performed the experiments, prepared figures and/or tables, and approved the final draft.

Juanzhen Zhao analyzed the data, prepared figures and/or tables, and approved the final draft.

Wenxue Jiang analyzed the data, authored or reviewed drafts of the article, and approved the final draft.

Xianjie Feng analyzed the data, prepared figures and/or tables, authored or reviewed drafts of the article, and approved the final draft.

Yuanting Li analyzed the data, prepared figures and/or tables, and approved the final draft.

The following information was supplied regarding data availability:

The raw data is available at figshare: Yang, Xue (2023): Spatio-temporal evolution and prediction of carbon storage in Kunming. figshare. Journal contribution. https://doi.org/10.6084/m9.figshare.22250713.v1.

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
