# Peer review of "Spatio-temporal evolution and prediction of carbon storage in Kunming based on PLUS and InVEST models"

_PeerJ, doi:10.7717/peerj.15285_

## Round 0.1 · original submission · Major Revisions

Dear Authors, as you shall see that the reviewers have now commented on your manuscript and are suggesting a major revision. Please revise your manuscript accordingly and incorporate all the suggestions made by them. I am looking forward to receiving your revised manuscript soon.

·

Basic reporting

This is a wonderful piece of research work well supported with nice dataset, analysis, result and discussion.
This should be accepted for the publication once authors will address following points:
1. In the abstract section, in line number 3, authors mention "carbon neutrality and peaking carbon emissions". However should peaking be replaced by reducing? Please check.
2. In the conclusion section, authors are suggested to add limitation of this study and way forward for future studies?

Experimental design

Overall this section is good. But authors are suggested to add more information on satellite images they have used for land change, like cloud cover, accuracy assessment etc. for better understanding.

Validity of the findings

This is perfectly done

Reviewer 2 ·

Basic reporting

The study used PLUS and InVEST models to estimate the of carbon storage in Kunming City of China and analyzed the impact of socio-economic factors and natural factors on carbon storage. The study is meaningful and uses a large amount of data, but there are problems such as disordered contents and insufficient in-depth analysis. The review result is major revision, which still needs further improvement and modification.

Experimental design

no comment

Validity of the findings

no comment

Additional comments

1. The language needs to be proofread and polished by a professional company or expert.
2. The prediction results of carbon storage lack comparison of similar studies, so it is difficult to determine whether the results are reasonable
3. There are a few errors, such as “base data” in Fig2, not “base date”.
4. Base data is lacking necessary descriptive statistical analysis, such as land use data in different periods, and driviing factors and data sources of Table2
5. In the discussion, the suggestions lack pertinence and should be combined with the actual in-depth analysis

Reviewer 3 ·

Basic reporting

You should define carbon storage. Is it the process of capturing and storing atmospheric carbon into biomass? Or is it the stock of carbon stored in biomass? The first definition is the same as carbon sequestration and about flow assessed during a time period, the second definition is the same as carbon stock and about stock assessed at a certain time point. Ecosystem services, in commonly accepted understanding, are flows.

Line 119: what do you mean “science foundation”? This is vague.

Introduction is too long. Description of Kunming in the introduction can be moved and combined into section 2.1 study area. Alternatively, Section 2.1 can be removed.

Line 170: are you actually assessing carbon sequestration or carbon storage? Please use consistent term.

The main models used in the study, including PLUS and INVEST, need to be explained. Authors should not assume readers already know what InVEST and PLUS are.

Line 216: what is Markov model.

Line 219: what are ecological construction goal

Line 237: what is Kappa co-efficient?

Line 393: what is Pearson analysis?

I believe the research results can benefit decision making in Kunming. However, since you’re trying to publish the paper on an international journal, you should justify how the paper can contribute to global context of carbon assessment.

Experimental design

Too many terms and acronyms that need to be explained.

Validity of the findings

How the findings can contribute to international carbon assessment should be justified

---

## Round 0.2 · Minor Revisions

Dear Authors,

Thank you for incorporating the suggestions made by the reviewers. Now kindly consider some minor revisions suggested by Reviewer 3.

Best regards
Gowhar Meraj

Reviewer 3 ·

Basic reporting

Line 94-95: what impacts do biodiversity and forests in Kunming have on the world? You should provide evidence.
Line 115-117: references needed.
Eco-civilization should be defined and explained.

Experimental design

looks ok

Validity of the findings

looks ok

---

## Round 0.3 · accepted · Accept

The authors have thoroughly revised their manuscript and it is now ready for publication. Thanks